# Discrepancies between FDA documents and ClinicalTrials.gov for Orphan Drug-related clinical trial data

**Mohua Chakraborty Choudhury**[¤❖], **Indraneel Chakraborty**[❖], **Gayatri Saberwal***

Institute of Bioinformatics and Applied Biotechnology, Bengaluru, India

❖ These authors contributed equally to this work.
¤ Current address: DST-Center for Policy Research, Indian Institute of Science, Bengaluru, India
* gayatri@ibab.ac.in

**Data Availability Statement:** This study was based on freely available data from the sources listed in this article. All other data generated or analyzed

## Abstract

Clinical trial registries such as ClinicalTrials.gov (CTG) hold large amounts of data regarding trials. Drugs for rare diseases are known as orphan drugs (ODs), and it is particularly important that trials for ODs are registered, and the data in the trial record are accurate. However, there may be discrepancies between trial-related data that were the basis for the approval of a drug, as available from Food and Drug Administration (FDA) documents such as the Medical Review, and the data in CTG. We performed an audit of FDA-approved ODs, comparing trial-related data on phase, enrollment, and enrollment attribute (anticipated or actual) in such FDA documents and in CTG. The Medical Reviews of 63 ODs listed 422 trials. We used study identifiers in the Medical Reviews to find matches with the trial ID number, 'Other ID' or 'Acronyms' in CTG, and identified 202 trials that were registered with CTG. In comparing the phase data from the 'Table of Clinical Studies' of the Medical Review, with the data in CTG, there were exact matches in only 75% of the cases. The enrollment matched only in 70% of the cases, and the enrollment attribute in 91% of the cases. A similar trend was found for the sub-set of pivotal trials. Going forward, for all trials listed in a registry, it is important to provide the trial ID in the Medical Review. This will ensure that all trials that are the basis of a drug approval can be swiftly and unambiguously identified in CTG. Also, there continue to be discrepancies in trial data between FDA documents and CTG. Data in the trial records in CTG need to be updated when relevant.

## Introduction

In order to provide a public record of each clinical trial, it is supposed to be registered in a registry that is accessible to the public. Such registries are rich sources of information about trials, and public registries make this information freely available to all users. The registration of clinical trials is important for several reasons. In particular, (a) it helps fulfill ethical obligations to trial participants; (b) since it is known that there is a tendency for sponsors to publish the results of those trials that have positive outcomes, but not others, the registration of each trial

during this study are included in this published article and the Supporting information files.

**Funding:** This work was partially supported by funds provided through a National Post-Doctoral Fellowship grant (2017–2019) (PDF/2016/002340) of the Science and Engineering Research Board of the Department of Science and Technology, Government of India to MCC. The project was also supported by internal funds of the Institute of Bioinformatics and Applied Biotechnology, which receives core support from the Department of Electronics, IT, BT and S&T of the Government of Karnataka. The funders had no role in the study whatsoever, including in the design of the study, data collection or analysis, preparation of the manuscript or decision on where to publish.

**Competing interests:** The authors declare that they have no competing interests.

ensures that the outcomes of every trial can be tracked. This reduces the publication bias that results from a selective publishing of trial results; and (c) it may increase enrollment in ongoing trials [1–3]. In addition, academic researchers and others have used such registries for several purposes such as (a) identifying trials that have not been in compliance with the law [4, 5]; (b) analyzing the trial activity of an institution over time [6]; (c) identifying the conditions for which trials have been conducted in a given country, relative to the local disease burden [7–9]; (d) analyzing the trial landscape for particular medical conditions [10] or technologies [11]; and (e) obtaining trial-related information that is unavailable in publications [12, 13]. Given the many potential uses of registry data, it is important to register every trial, and ensure that the data in the trial record are comprehensive and accurate.

ClinicalTrials.gov (CTG) is the public, United States (US) government trial registry. It has become the major registry in the world [14], with the largest number of records [15], almost 150,000 unique visitors every day [16] and about 215 million page views per month [17]. Created as a result of the Food and Drug Administration Modernization Act of 1997, it has been accessible to the public since February 2000. The International Committee of Medical Journal Editors was an early mover in the effort to pressure researchers to register their studies in such a registry [3], and since then regulators [18], funders [19], ethics committees [20], clinician scientists [21], academic researchers [22] and others have added their voices to this endeavor. Despite such efforts, many trials are still not registered [23–25]. The US government's Clinical Trials Registration and Results Information Submission (Final Rule) of 2017 has re-emphasized the need to register trials, and all the stakeholders who are concerned about the integrity of the trial enterprise hope that this will ensure compliance [26].

In this study we have focused on trials of drugs for diseases that affect very few people and are collectively known as rare diseases (RDs). Each country has a different threshold, and in the US a condition or diseases that affects fewer than 200,000 people in the country is identified as an RD. Drugs for such diseases, or conditions, are known as orphan drugs (ODs). Examples of ODs are Zolgensma for Spinal Muscular Atrophy, Exondys 51 for Duchenne Muscular Dystrophy, Idursulfase for hunter syndrome, Velaglucerase Alpha for gaucher disease and Agalsidase Alfa for fabry disease.

At the R&D stage, the FDA grants a special 'orphan drug designation' for such drugs, that qualifies them for special incentives including R&D tax benefits and seven years' market exclusivity. Conducting clinical trials for RDs can be a challenge given that the number of patients is small, and the patients are often geographically dispersed. Unlike diseases that affect a larger population, often there are insufficient numbers of RD patients to participate in different phases of trials, to generate a statistically significant outcome [27]. In addition, most RDs are severe, debilitating conditions which makes it difficult for patients to participate in a trial [28–30]. Finally, RDs are often not well understood due to insufficient data related to clinical characterics of the disease and the lack of natural history studies [31]. Thus, in the case of ODs, comprehensive and updated trial information is even more important to strengthen the research base. It is therefore essential that OD studies are registered, and the data in the trial record are complete, accurate and kept up to date.

Aside from a trial registry, the Food and Drug Administration (FDA) also receives details of all trials related to the candidate drugs it is evaluating. Subsequently, the FDA makes freely available certain documents that contain trial data underlying the approved drug.

In this study we have performed a limited audit of trial data. For trials related to certain FDA-approved ODs, we asked whether there were discrepancies in certain fields of data in the FDA and CTG. If so, we wished to quantify the discrepancies. This study identifies and quantifies the gaps–for three parameters–in the trial record of a set of approved ODs.

**Table 1. Sources of Orphan Drug-related clinical trial data.**

| Database | What it contains | Why we needed this data |
|---|---|---|
| **FDA's Orphan Drug Designations and Approvals database (FDAOD)** | Lists all FDA-approved Orphan Drug product designations and approvals | To identify the list of Orphan Drugs approved between 1 January 1983 and 17 April 2020 |
| **Orange Book** | Catalogues all the small molecule drugs ever approved by the FDA | Looked for matches for 413 drugs listed in FDAOD, that had a single approval |
| **Drugs@FDA** | Includes information about all drugs approved for human use in the US | Searched for Clinical Review, the Multidiscipline Review or the Medical Review of each drug of interest |
| **ClinicalTrials.gov (CTG)** | Is a database of clinical trials and other clinical studies | Searched for clinical trial information of trials related to the 63 Orphan Drugs. |

## Materials and methods

This was a retrospective, descriptive study based on publicly available information from the websites of the US FDA and CTG, with the relevant links available as references [32–35]. All the data were initially collated by MCC, and verified by one or both of the other authors. Discrepancies were resolved by discussion. We obtained data from multiple sources, summarized in Table 1, and described below.

### Defining the dataset

We took several steps to identify the ODs of interest, summarized in Fig 1.

On 18 April 2020 we accessed the FDA's Orphan Drug Designations and Approvals database (FDAOD) [32], and downloaded the list of orphan drugs (ODs) that had received marketing approval in the US between 1 January 1983 and 17 April 2020, inclusive. This resulted in an excel file with details of 862 approvals. 40 drugs did not list a trade name. The rest of the approved drugs had a total of 552 unique trade names. Based on the number of orphan approvals of each drug, the dataset was divided into (i) 413 trade names that had received a single approval, and (ii) 139 that had received multiple approvals (S1 File). For each drug, the FDAOD provided the generic name, trade name, marketing approval date, and indication (S2 File).

We wished to study drugs that had received marketing approval for an orphan indication under a single New Drug Application (NDA), and had not received approval for an orphan- or non-orphan indication under another NDA. To identify these cases, we compared our single approval list with the list of drugs in the Orange Book, which catalogues all the small molecule drugs ever approved by the FDA, whether for orphan- or non-orphan indications. We downloaded the data files from the Orange Book Database [33] on 18 April 2020. The download was in the form of a compressed data folder that contained three files, viz. exclusivity.txt, patent.txt and products.txt. We compared the single approval list with the data in products.txt to determine the number of approvals per drug. 123 of the 413 single approval drugs–as listed in the FDAOD–were listed in the Orange Book only once, and therefore qualified for this study (S1 File). To be noted, since the Orange Book does not list biological products [34] our dataset did not include any biologic.

Of the 123 drugs, we chose to study the ODs approved after 2008, ie. from January 1st, 2009–April 17th, 2020, inclusive (S1 File). For each of these 72 drugs, we searched the Drugs@FDA site [35] for the availability of the Clinical Review, the Multidiscipline Review or the Medical Review. For simplicity we refer to all of these documents as MedRs. Only 63 of the 72 drugs had a publicly available MedR, and the FDA documents of the 63 ODs were used with the following frequencies: Clinical Review (12), Multidiscipline Review (15), and Medical Review (36) (S1 File). It turned out that 27 (43%) of these ODs were approved between 2009

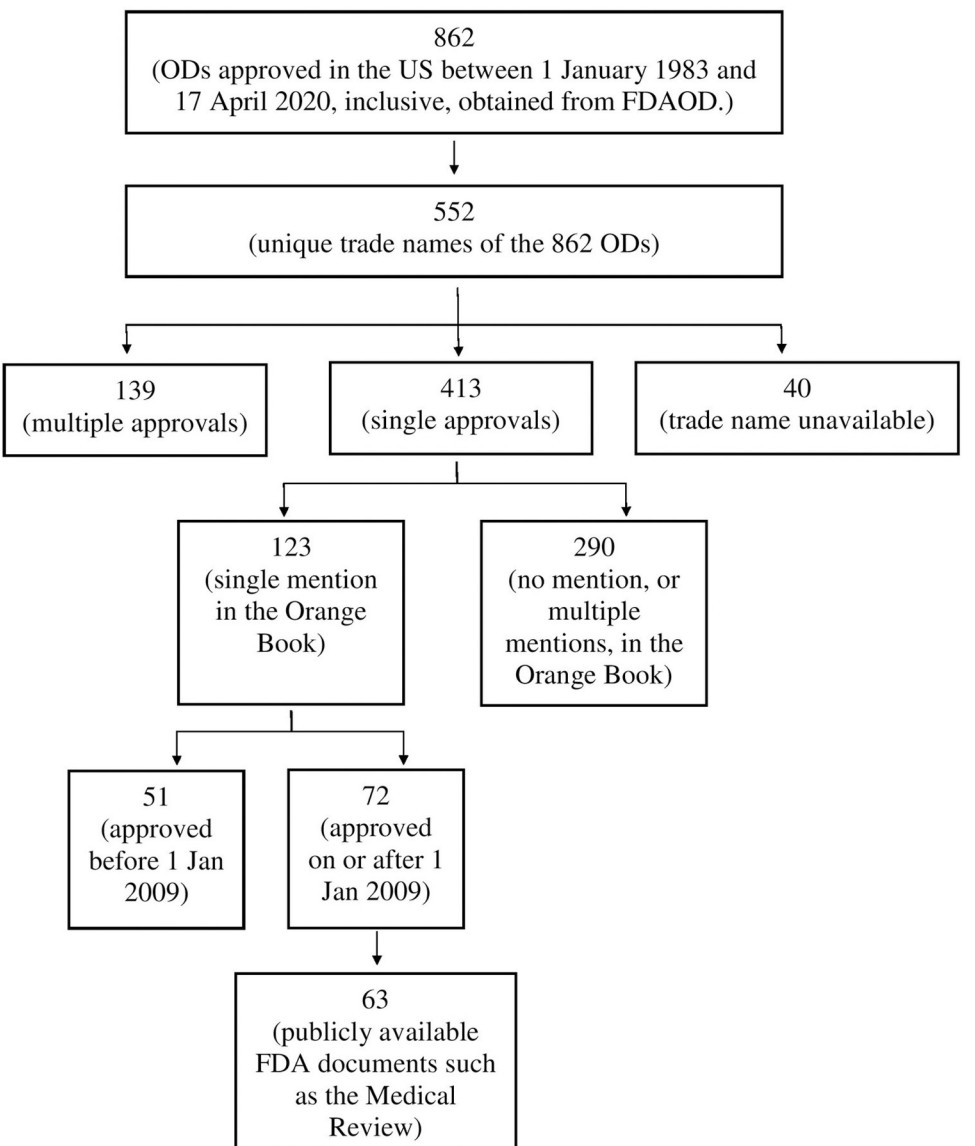

**Fig 1. The steps involved in identifying the 63 ODs for this study.**

and 2014, inclusive, and 36 (57%) between 2015 and 2019, inclusive (S1 File). As such, the data analyzed in this study tended to be from more recently approved ODs.

In summary, our final dataset consisted of 63 ODs, approved after 2008. Each drug (i) was listed once each in FDAOD and OB, (i) was approved for a single indication, and (iii) had a publicly available MedR.

## Data extraction from the FDA MedRs

We identified the indication for which approval was sought by examining the first few pages of each of the MedRs. Scripts were used for preliminary work. All final data was both extracted and verified manually. To identify trials (and we have used the terms 'trial' and 'study' interchangeably) related to this indication we examined the table with a title such as 'Table of Clinical Studies' or 'Tables of Studies/Clinical Trials' (henceforth, the Table), which is part of the

section 'Sources of Clinical Data and Review Strategy'. For most drugs, all the information related to the clinical studies that led to FDA approval was available from this section.

Pivotal trials are the crucial trials containing proof of efficacy of the drug molecule. These are among the most important trials whose data is submitted to the FDA as part of a new drug application. Therefore, we went on to search the entire MedR for the word 'pivotal', in order to annotate the relevant trial with the 'pivotal' descriptor if this was missing in the Table. In the case of drug Brukinsa, study BGB-3111-306 was not listed in the Table at all, and we added it to the list of pivotal trials.

There were some exceptions to this protocol: (a) Some of the MedRs did not list the Table. In such cases, we searched the entire MedR for relevant trials. (b) In the case of Firdapse we did not consider a safety study that lacked an ID. (c) In the case of Galafold, we did not consider the six Phase 2 trials and 10 Phase 1 trials, that lacked trial IDs. For each drug, the relevant MedR supplied the following details of each trial: the study identifier; the pivotal status, if relevant; phase; enrollment (which we termed N1); and enrollment attribute (S2 File). We term the enrollment data from the MedR document as N1.

## Data extraction from CTG

Next, for the 63 drugs, we captured clinical trial information from CTG. CTG data was extracted via CTG's API using rclinicaltrials package, currently archived in CRAN, in R scripting. We did this for the indication for which each had received approval as an OD. CTG was searched in three steps using three different identifiers: Trade name of the OD, generic name of the OD and study identifiers of the clinical trials. These identifiers were first searched in CTG using a python program and later double checked manually for accuracy. A match for the study identifier was sometimes found in the 'Other ID' or 'Acronym' field of the CTG record. For all the identified studies, data from the following fields were extracted from the CTG record: NCT number; acronym; other IDs; condition; phase; enrollment; and enrollment attribute (S2 File). We term the enrollment data from CTG as N2. Sometimes N1, in the MedR, was available as both planned and actual enrollment. Since N2 had only one attribute, either planned or actual, we considered whichever N2 attribute was available, to compare with the attribute of N1.

For each OD, the data on studies mentioned in the MedR, and that obtained from the corresponding ones in CTG, if available, were collated in an excel sheet (S2 File). The trials that matched were presented in the same row for easy comparison. We went on to compare the two sets of data, trial-wise. For a given drug, we examined whether each relevant trial listed in the FDA document was also registered with CTG. If so, we compared details of (i) phase, (ii) enrollment, and (iii) enrollment attribute ('anticipated' or 'actual') from the two sources. We did this for all trials, and separately for the sub-set of pivotal trials.

## Results

From the MedRs, we extracted data for 422 studies associated with the 63 ODs (S2 File). In searching for these trials in CTG, the trade names of these drugs helped identify 151, the generic names 47, and the study identifiers, four trials (S2 File). This totaled to 202 studies, which we termed MedR-CTG pairs. CTG matches were not found for the remaining 220 studies. These 220 studies could not be analyzed, and were filtered out.

### A comparison of the data in MedR and CTG

For the 202 MedR-CTG pairs, we examined the (i) phase, (ii) enrollment, and (iii) enrollment attribute ('anticipated' or 'actual'). The attribute is important primarily as a qualifier of the enrollment. That is, it ensures that the same enrollment is being compared.

**Table 2. For 202 trials, a comparison of the phase information in the FDA's MedR documents and in Clinical-Trials.gov.**

| No. | Relationship of phase in MedR and CTG | Number of trials | Percentage of trials |
|---|---|---|---|
| 1 | An exact match of CTG data with that from the MedR Table | 152 | 75.2 |
| 2 | An exact match after a manual search of the entire MedR | 11 | 5.4 |
| 3 | An overlap, such as Phase 3 in the MedR Table and Phase 2/3 in CTG | 18 | 8.9 |
| 4 | A blank in CTG. These were Expanded Access studies, which matched the information in the MedR. | 3 | 1.5 |
| 5 | A given study's phase was not described anywhere in the MedR | 14 | 6.9 |
| 6 | We did not search for information in the MedR because it was not searchable by a Ctrl+F function | 2 | 1.0 |
| 7 | Discrepancies in the MedR Table and CTG | 2 | 1.0 |
| | **Total** | **202** | **100** |

Abbreviations: **CTG**: ClinicalTrials.gov.; **MedR**: the Clinical Review, the Multidiscipline Review or the Medical Review document of the FDA, containing data related to a give drug's approval.; **Table**: the summary table with a title such as 'Table of Clinical Studies' or 'Tables of Studies/Clinical Trials', which is part of the section 'Sources of Clinical Data and Review Strategy' of a MedR.

**Phase.** For the phase, the 202 MedR-CTG pairs fell into seven categories as described in Table 2 and S3 File. An exact match of CTG data with that from the MedR Table comprised the largest fraction, with 152 cases. However these comprise only 75.2% of the trials.

**Enrollment.** As mentioned, we used N1 to denote the enrollment listed in the MedR, and N2 that in CTG. Of the 202 MedR-CTG pairs, the study was listed as ongoing in the MedR in two cases, and therefore N1 was not a final figure. In 10 (5%) studies, either N1 or N2 (5 cases each) was missing. We compared N1 and N2 of the remaining 190 studies (Table 3 and S4 File).

For 132 (69.5 of 190) studies there was an exact match (that is, N1 = N2) and in 25 (13.2% and 12.4%, respectively) studies, N2 was within ± 5% of N1. Together, these 157 (82.6% and 77.7%, respectively) pairs were referred to as Closely Matched Trials. For 33 (17.4% and 16.3%, respectively) studies, N2 was not within 5% of N1. If we expanded the window from +/− 5% to +/− 10%, a mere 4 studies were now included in the Closely Matched Trials.

**Attribute of enrollment.** In 173 (91.1% of 190 and 85.6% of 202) studies, the attributes of N1 and N2 matched, and in 17 (8.9% and 8.4%, respectively), they did not (S4 File). Of the 17,

**Table 3. For the trials present both in the FDA's MedR documents and in ClinicalTrials.gov (the MedR-CTG pairs), a comparison of the enrollment in the Med R (N1) and ClinicalTrials.gov (N2).**

| No. | Relationship of N1 and N2 | Number | Percentage of MedR-CTG pairs |
|---|---|---|---|
| 1 | N1 = N2 | 132 | 69.5 |
| 2 | N2 within + 5% of N1 | 17 | 8.9 |
| 3 | N2 within− 5% of N1 | 8 | 4.2 |
| 4 | N2 not within + 5% of N1 | 22 | 11.6 |
| 5 | N2 not within− 5% of N1 | 11 | 5.8 |
| | **Total** | **190** | **100** |

The frequency of cases (a) of N1 = N2, (b) of N2 being within + 5% of N1, (c) of N2 being within− 5% of N1 (d) of N2 not being within + 5% of N1, and (e) of N2 not being within− 5% of N1.

Abbreviations: **MedR-CTG pair**: Trials for which data from both MedR and CTG were available.; **N1**: the enrollment data from the MedR document.; **N2**: the enrollment data from CTG.

(i) eight had no attribute in CTG, and (ii) for nine, N2 was listed as 'anticipated', compared to the 'enrolled' status of N1.

## A comparison of the data in MedR and CTG, for pivotal trials

In the MedRs, some trials were marked pivotal, key, primary or important. For 43 ODs, one or more studies were so marked, whereas for 20, none were (S5 File). Overall, 82 studies were so described (S5 File), and we termed all such trials as pivotal. We examined this sub-set separately for phase, for enrollment, and for the attribute of enrollment.

Of the 82 trials, for 30 (36.5%), the CTG match could not be identified. However, N1 was not available for one study, so we proceeded to analyze 51 (62% of 82) trials. These 51 studies were a subset of the 190 MedR-CTG pairs.

**Phase.** As for the larger set of trials, above, we compared the phase information for each pivotal study in the MedR and in CTG. The trials fell into four of the seven categories as described in Table 4 and S5 File. An exact match of CTG data with that from the MedR Table comprised the largest fraction, with 44 (86.3% of 51) cases.

**Enrollment.** Of the 51 MedR-CTG pairs, in 34 (67% of 51) cases, N1 matched N2 (Table 5, S5 File), but in 17 (33%) studies, it did not. Among the 17, in eight (47.1% of 17) studies, N2 was a Closely Matched Trial, whereas in nine (52.9%), it was not. If we expanded the window from +/- 5% to +/- 10%, no further studies were included in the Closely Matched Trials.

**Attribute of enrolment.** Of the 51 studies, for 48 (94.1%), the attribute of N1 matched that of N2, and for three (5.9%) it did not (S5 File).

## Discussion

For a well defined set of recently approved ODs, we compared trial-related data on phase, enrollment, and enrollment attribute (anticipated or actual) in FDA documents and in CTG. Although a comparison of phase and enrollment may be more obvious, there were two reasons why we assessed the enrollment attribute. First, because the attribute in the MedR and CTG

**Table 4. For 51 pivotal trials, a comparison of the phase information in the FDA's MedR documents and in ClinicalTrials.gov (CTG).**

| No. | Relationship of phase in MedR and CTG | Number of trials | Percentage of trials |
|---|---|---|---|
| 1 | An exact match of CTG data with that from the MedR Table | 44 | 86.3 |
| 2 | An exact match after a manual search of the entire MedR | 2 | 3.9 |
| 3 | An overlap, such as Phase 2b in the MedR Table and Phase 2 in CTG | 3 | 5.9 |
| 4 | A given study's phase was not described anywhere in the MedR | 2 | 3.9 |
| | **Total** | **51** | **100** |

**Table 5. For the 51 pivotal trials, a comparison of the enrollment data from the MedR document (N1) and ClinicalTrials.gov (N2).**

| No. | Relationship of N1 and N2 | Number | Percentage |
|---|---|---|---|
| 1 | N1 = N2 | 34 | 66.7 |
| 2 | N2 within + 5% of N1 | 5 | 9.8 |
| 3 | N2 within– 5% of N1 | 3 | 5.9 |
| 4 | N2 not within + 5% of N1 | 7 | 13.7 |
| 5 | N2 not within– 5% of N1 | 2 | 3.9 |
| | **Total** | **51** | **100** |

needed to match if a meaningful comparison of N1 and N2 was to be made. Second, because CTG needs to be kept updated. It is possible that the final data concerning a trial was submitted to the FDA, without the necessary changes being made in CTG. Outdated information is misleading for those stakeholders who seek information about a trial from CTG, and needs to be pointed out in studies such as this one.

We chose to work with drugs approved after 2008 due to the regulations in force at the time. The United States Food and Drug Administration (FDA) Amendments Act of 2007 (U. S. Public Law 110–85), was signed into law on 27 September 2007 [36]. This Law mandated the registration of 'applicable clinical trials' [37], that is studies–other than Phase 1 trials–used to support the application to the FDA for a new drug's approval, in a publicly accessible trial registry. Although the law came into force in September 2007, we left ample margin, and examined drugs approved after 2008, to maximize the number of drugs whose trials would be available in ClinicalTrials.gov.

Further, we chose to work with a well defined set of ODs, that is those that had been approved for only one orphan indication, and had not been approved for any non-orphan condition. Finally, over time, there has been an improvement in the rates of registration of trials in public registries [38, 39]. This should lead to the identification of more MedR-CTG pairs.

We needed to use three identifiers–the trade name of the OD, the generic name of the OD and the study identifier of a given study–listed in the MedR, to identify trials' counterparts in CTG. Even so, in 52% of the cases, we could not find a match. There are various possible reasons why we could not find a trial in CTG. (i) the studies may have run before CTG was established. (ii) Phase 1 trials were not required to be registered until the Final Rule of 2017. (iii) In the United Kingdom (UK), an audit has found that 12% of trials have not been registered [25] and a certain fraction of trials may remain unregistered in the US as well. (iv) It is possible that we were unable to identify the CTG match of some studies although such a match did exist. As argued by others as well [40], in the interest of transparency, it is important to provide the NCT IDs of all studies listed in a MedR. This will ensure that all trials that are the basis of a drug approval can be swiftly and unambiguously identified in CTG.

Summarizing our findings, by focusing on the 'Table of Clinical Studies' of the Medical Review, and comparing it with the data in CTG, for phase there was an exact match in only 75% of the cases. The enrollment matched only in 70% of the cases, and the enrollment attribute in 91% of the cases. A similar trend was found for the sub-set of pivotal trials, where 86% of phase, 67% of enrollment and 94% of enrollment attribute matched.

We now discuss the quality of the data pertaining to studies listed in both the MedR and in CTG. Of the 202 MedR-CTG pairs, N1 or N2 was missing in 10 of them. This data ought not to be missing, since most of the trials were completed years ago. Of the 190 pairs that were analyzed, N1 from the MedR Table matched N2 only in 70% of the studies. It is surprising that there is a discrepancy in such a large fraction of trials. Even on relaxing the value to within 5% of each other, there were discrepancies in 17% of the trials. This is a substantial number. Many individuals and organizations have, for many years, stressed the need for (i) all trials to be registered [2, 3, 41], (ii) all data in registries to be accurate [23, 42], (iii) the results of trials to be publicly declared, on time [43, 44] and (iv) those results to be accurate [13, 45]. The importance of these issues is such that the UK House of Commons published a report reiterating the need for these steps [46]. If N1 and N2 –which are numbers describing the enrollment in the same trial–do not match, it means that data either in the registry or in FDA documents is erroneous. Both the registry and the FDA are government organizations, and should host correct data. If there are discrepancies in such a simple number, it throws into doubt the veracity of other information in the registry or submitted to the FDA.

In the case of the attributes of N1 and N2, the MedR and CTG records were in much better agreement, since only 17 (9%) of 190 pairs were different. The studies with missing N2 attributes were of drugs approved from 2009–2012, and studies whose N2 status was not updated were of drugs approved from 2009–2019. Although 9% is a relatively low figure, CTG data is supposed to be kept up to date, and therefore there should be no discrepancies. Separately, we examined the pivotal trials, that is the most important studies supporting the application for the approval of a drug candidate. It is surprising that such a large fraction– 36.5%–of pivotal trials were not registered with CTG. In general, the records of these studies were no more accurate than those of the entire set of trials for which MedR-CTG pairs existed. The discrepancies in N1 and N2 are likely due to the sponsors not ensuring that registry data is correct and up-to-date.

Information about clinical trials is primarily sought from trial registries. Although it is important that all studies are registered, it is also important that the data in each record are accurate. Registry data are not peer reviewed [47], and it is known that there are many types of errors in the data in CTG and other public registries [24, 42, 48, 49], or discrepancies in the data of a particular study listed in multiple registries [50]. Aside from registries, researchers have used academic publications and regulatory documents to obtain trial-related information, and discrepancies have been identified when data in (i) a registry and the publication [51–53]; (ii) a registry and FDA documents [13]; and (iii) a registry, the publication and the FDA documents [54] were compared.

As mentioned earlier, the International Committee of Medical Journal Editors and other organizations, and individuals, have pressured trialists to register and report their studies. Nevertheless, problems of non-registration, non-reporting of trial results (in a timely manner), and discrepancies in trial-related data persist. As a result, there have been various initiatives, or suggestions for initiatives, that would increase confidence in trial-related data, as exemplified by the following: (i) Academic researchers and certain health-related organizations have repeatedly called for audits [44, 55]; (ii) In 2018, the Science and Technology Committee of the UK House of Commons recommended that the Health Research Authority audit all trials [46]; (iii) AllTrials.net has named and shamed those who have not reported their results in a timely manner [56]; (iv) The OpenTrials initiative intends to host all the publicly available information on a given study, from registries, publications, regulatory documents and so on [57]; and (v) An audit of the policies of major philanthropic and government funders of medical research has been conducted, to determine their requirements of trial result reporting [58].

Although the quality of data in CTG has received considerable attention, FDA documents have not been analyzed as much. This study reinforces the idea that, periodically, trial data from multiple sources must be compared to ensure consistency. As a first step, the OpenTrials initiative aims to collect all publicly available information that it can locate, for every trial that has been conducted, around the world. A comprehensive repository of this sort would greatly facilitate audits. The automation of such audits, using artificial intelligence and machine learning would also be greatly facilitated by the organization of data in each document type using templates. The older FDA documents are compilations of scanned images and hence not easily machine readable. Even in newer FDA documents, the contents of the summary 'Table of Clinical Studies' shows a great deal of variability from one document to the next.

## Limitations

This study has several limitations. (i) The MedR contains a comprehensive summary of the clinical trial data submitted as part of an NDA. This document tends to be hundreds of pages long [59]. A given MedR may also be available only as multiple shorter documents, and the

documents related to older approvals may be in a scanned format and challenging to read. Therefore, we limited our dataset to more recently approved ODs, and this made the task of examining the MedRs easier. (ii) It is based on a small set of ODs, and it is not clear whether the results are generalizable to other ODs or non-ODs. (iii) The small set of studies, linked to these ODs, was further truncated due to various lacunae in the data. (iv) Also, the study examined the discrepancies between the MedR and the CTG record of a small set of fields, and the extent of discrepancies in other fields is unknown.

## Conclusions

We examined the pivotal, and other, trials underlying the approval of a well-defined set of ODs. We compared the data related to phase, to enrollment and to enrollment attribute in the FDA MedR document versus that in CTG, and quantified the discrepancies. From the point of view of the accountability of the trial enterprise, especially important for ODs, for which the patient base is small, these discrepancies need to be done away with. It needs to be continually emphasized that CTG data need to be kept up to date for a given trial. It is also important to provide the NCT IDs of all studies listed in a MedR, to facilitate audits.

## Supporting information

**S1 File. The steps involved in identifying the 63 ODs for this study. a**. The list of 862 ODs approved in the US between 1 January 1983 and 17 April 2020, inclusive, obtained from FDAOD. The information available in FDAOD is provided. **b**. The list of the 862 ODs' 552 unique trade names, and their break up into (i) 413 with single approvals, (ii) 139 with multiple approvals and (iii) 40 cases with trade name unavailable. **c**. Of the 413 single approval trade names, 123 had a single mention in the OB, and 290 either did not have any mention or had multiple mentions in the OB. **d**. Of the 123 ODs with a single occurrence in the OB, 51 were approved before 1 Jan 2009, and 72 after. **e**. Of the 72 ODs approved on or after 1 Jan 2009, 63 had publicly available comprehensive FDA documents such as the Medical Review. **f**. The list of 63 ODs of this study, with their year of marketing approval and the FDA document consulted for this study. **g**. From 2009–2020, per year (i) the total number of ODs approved, (ii) the number selected for this study, (iii) the percentage selected for this study, (iv) the percentage out of 63 ODs, (v) the cumulative percentage out of 63 ODs, and (vi) the cumulative percentage, out of 63 ODs, in three year segments.
(XLSX)

**S2 File. The list of matches found in CTG using the trade name, the generic name, or the trial study identifier, and a summary of this list; details of each trial from the MedR and CTG; and the sources of information for 63 ODs and their trials. a**. For the 63 ODs, sources of particular drug and trial information. **b**. For the 63 ODs, data extracted for each relevant trial from the MedRs, and from CTG where possible. **c(i)**. For trials listed in the MedR, the list of matches found in CTG using the trade name, the generic name, or the trial study identifier. **c(ii)**. Summary of the frequency with which matches were found in CTG using the trade name, the generic name, or the trial study identifier.
(XLSX)

**S3 File. A comparison of the phase of 202 trials, as listed in the MedR and in CTG.** a. An exact match, as deduced from the Clinical Trials Table. b. An exact match after a manual search of the entire MedR. c. An overlap, such as Phase 2b in CTG but Phase 2 in CTG or Phase 3 in CTG and Phase 2/3 in CTG. d. A blank in CTG, and these were Expanded Access cases, which matched the data in MedR. e. The word 'phase' was not linked to a given trial

anywhere in the MedR. f. We did not search for information due to the non-searchability of the MedRs. g. Clear discrepancies.
(XLSX)

**S4 File. For the MedR-CTG pairs, a comparison of the N1 and N2 values and attributes. a (i)**. A list of the 12 MedR-CTG pairs of trials, where the value of either N1 or N2 was unavailable. **a(ii)**. A comparison of the N1 and N2 values for the relevant MedR-CTG pairs: (a) The 190 MedR-CTG pairs with N1 and N2 values available; (b) N1; (c) N1+5%; (d) N1-5%; (e) N2; (f) trials where N1 = N2; and trials where N2 was within +/- 5% of N1. **b**. For 190 MedR-CTG pairs, (i) N1, (ii) N1 attribute, (iii) N2, (iv) N2 attribute and (v) whether or not the two attributes match.
(XLS)

**S5 File.** (i). A listing of the MedR-CTG pairs of pivotal trials; and for these pairs, a comparison of the phase, and of the N1 and N2 values and attributes. a. A list of the ODs that did, or did not, list pivotal trials. b. The trials that were marked as pivotal, important, key or primary, that (i) did have CTG matches, (ii) that did have a CTG match, but for which N2 was unavailable and (iii) did not have CTG matches. c. A comparison of the N1 and N2 values for the 51 pivotal trials (a) The 51 pivotal trials with N1 and N2 values available; (b) N1; (c) N1+5%; (d) N1–5%; (e) N2; (f) trials where N1 = N2; and (g) trials where N2 was within +/- N1 and 5%. d. For 51 pivotal trials, (i) N1, (ii) N1 attribute, (iii) N2, (iv) N2 attribute and (v) whether or not the two attributes match. (ii). A comparison of the phase of each of 51 pivotal trials, as listed in the MedR and in CTG. e. An exact match, as deduced from the Clinical Trials Table. f. An exact match after a manual search of the entire MedR. g. An overlap, such as Phase 2b in CTG but Phase 2 in CTG or Phase 3 in CTG and Phase 2/3 in CTG. h. The word 'phase' was not linked to a given trial anywhere in the MedR.
(XLS)

## Acknowledgments

We thank Mounika Pillamarapu for assistance in the very early stages of this project, and Jaishree Mendiratta for later help.

## Author Contributions

**Conceptualization:** Gayatri Saberwal.

**Formal analysis:** Mohua Chakraborty Choudhury, Indraneel Chakraborty, Gayatri Saberwal.

**Funding acquisition:** Mohua Chakraborty Choudhury, Gayatri Saberwal.

**Investigation:** Mohua Chakraborty Choudhury, Indraneel Chakraborty.

**Methodology:** Mohua Chakraborty Choudhury, Gayatri Saberwal.

**Project administration:** Gayatri Saberwal.

**Resources:** Gayatri Saberwal.

**Software:** Mohua Chakraborty Choudhury, Indraneel Chakraborty.

**Supervision:** Gayatri Saberwal.

**Validation:** Indraneel Chakraborty, Gayatri Saberwal.

**Writing – original draft:** Mohua Chakraborty Choudhury, Gayatri Saberwal.

**Writing – review & editing:** Mohua Chakraborty Choudhury, Indraneel Chakraborty, Gayatri Saberwal.

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
