## [Decision Letter · Decision Letter 0]

21 Oct 2021

PGPH-D-21-00617

For orphan-drug related clinical trial data, discrepancies between FDA documents and ClinicalTrials.gov

Dear Dr. Saberwal,

Thank you for submitting your manuscript to PLOS Global Public Health. After careful consideration, we feel that it has merit but does not fully meet PLOS Global Public Health’s publication criteria as it currently stands. Therefore, we invite you to submit a revised version of the manuscript that addresses the points raised during the review process.

We look forward to receiving your revised manuscript.

Kind regards,

Genevieve Cecilia Aryeetey, Ph.D

Academic Editor

Journal Requirements:

1. Please ensure that the Title in your manuscript file and the Title provided in your online submission form are the same.

2. We have noticed that you have uploaded supporting information but you have not included a list of legends.  Please add a full list of legends for all supporting information files (including figures, table and data files) after the references list. 

Additional Editor Comments (if provided):

Refer to comments from all 4 reviewers

Reviewers' comments:

Reviewer's Responses to Questions

**Comments to the Author**

1. Does this manuscript meet PLOS Global Public Health’s publication criteria? Is the manuscript technically sound, and do the data support the conclusions? The manuscript must describe methodologically and ethically rigorous research with conclusions that are appropriately drawn based on the data presented.

Reviewer #1: Partly

Reviewer #2: Partly

Reviewer #3: Yes

Reviewer #4: Yes

2. Has the statistical analysis been performed appropriately and rigorously?

Reviewer #1: N/A

Reviewer #2: N/A

Reviewer #3: N/A

Reviewer #4: No

3. Have the authors made all data underlying the findings in their manuscript fully available (please refer to the Data Availability Statement at the start of the manuscript PDF file)?

Reviewer #1: Yes

Reviewer #2: Yes

Reviewer #3: Yes

Reviewer #4: Yes

4. Is the manuscript presented in an intelligible fashion and written in standard English?

Reviewer #1: Yes

Reviewer #2: Yes

Reviewer #3: Yes

Reviewer #4: Yes

5. Review Comments to the Author

Reviewer #1: The authors presented a descriptive study that compared certain information on orphan drug clinical trials between clinicaltrial.gov and FDA registries. The authors found that there are some discrepancies between the information available in these databases. Although there are other published papers that reported similar findings, those are related to clinical trials of drugs for other diseases or compared different types of information. This is a work that can support the need to improve the quality of information registered with Clinicaltrial.gov and FDA. In general, the manuscript is understandable, however, there is information that is not completely clear, and it would be helpful to reorganize some descriptions to improve the quality of the work.

Major issues

In Materials and Methods, the authors presented a very detailed description for the selection of the registries to evaluate, including numbers. However, this may be better located as part of the results.

Although the authors did a good job explaining the study methodology, adding some information would be helpful to better understand the work.

For example, how exactly they did the comparison between different files, if it was by software or manually.

Why they chose studies approved after 2008.

The reasons for limiting the number of diseases or trials to review the registries and some definitions like “pivotal studies”, these two aspects are presented in the discussion, however, it would be useful to the readers to know them since the methods.

A more detailed definition of N1 and N2.

Minor issues

While the abstract provides the reader with an overview of the work, some information about the methodology is missing, specifically how the comparison was made between the different registries.

As there was a long cascade to choose the final registries to review, it would be useful to the reader to have a flow chart with the numbers that lead to the final number, showing the selection criteria at each stage.

Although the intention of the tables is to show the findings in a comprehensive manner, tables 2 and 4 do not add much to the description made by the authors in results section. These descriptions can be summarized by presenting the most relevant results.

The description of table 2 began before the table was placed, however after the table there is a complementary description, with all the information included in the table. This description could be summarized and presented before the table to give a better flow of the information presented.

It would be useful to the readers if the authors include the description of N1 and N2 in the table captions, as well as the description of abbreviations used in the tables.

Reviewer #2: I have put my comments in the attached file. I hope the authors find them useful. Most of my comments have to do with clarity and justification for various decisions the authors have made (e.g., the date range, the outcomes). I think the fact the authors could not find CTG data for 200 of the 422 trials is an important one, even if the attributes were to match closely for the 202 trials. In Tables 2 and 5, it would be better to break down rows 2 and 3 (the ones about N1 and N2) into N1>N2 and N1<n2, authors="" currently="" discussion="" have.="" in="" instead="" of="" the="" then="" what=""></n2,>

Reviewer #3: Overall it’s a well written manuscript highlighting an important area of quality assurance and human safety for clinical trials in general and orphan drugs in particular. A lot of hard work and efforts have gone in conducting this audit.

Below are few suggestions to further strengthen the manuscript:

1. Though the authors have defined the term “rare disease”, but would be better if they can also share few examples.

2. Under the discussion section, the authors have highlighted “Although the quality of data in CTG has received considerable attention, FDA documents have not been analyzed as much. This study reinforces the idea that, periodically, trial data from multiple 306 sources must be compared to ensure consistency.” The authors could also comment on how this exercise and audit could be streamlined through the use of technology and artificial intelligence. For instance, development of databases which can collate information from different sources, prepares the audit report and highlight the gaps.

Reviewer #4: General comments:

The paragraphs don't flow together very well, so I suggest trying to link them together a bit better. The introduction section is a bit disconnected. I would also suggest having a colleague (who isn't working on this project) proofread to help improve how the text reads, since it is a bit hard to follow in places.

I think some terms need defining more explicitly in the text for those who aren't completely familiar with the language of trials.

There are too many acronyms for the text to be easily readable, so I suggest spelling some of them out that are actual expressions, like study identifier , orphan drug, or rare disease. I would also move the abbreviations section above the references.

I would also suggest introducing your research aim/question more explicitly, and also putting the 'why' in here (same paragraph). Then refer back to this in the conclusions, since it's not very clear what question you answered.

Further, I think the materials and methods section needs to be revised to make clearer what you did. S1 is quite clear, so I suggest making this into a flow diagram and making it figure 1. For the text I suggest the following order:

1. Which datasets did you obtain? What did they contain, in terms of variables and why did you need them? (i.e. FDAOD, New Drug Application, Drugs@FDA (?), Orange Book?, MedRs, CTG (?))

2. How did you extract the data? If it was with a script, what was the script scripted in, and how did it work? Inclusion/exclusion criteria here.

3. How did you combine the data together? If into one dataset, how did this dataset look? A table of variables/structure of the final dataset could help here.

To improve the paper, I would suggest doing some simple statistical tests to compare the pivotal trials and all trial datasets. However, the pivotal trials are contained in the all trials dataset (as far as I understand), so you cannot easily draw comparisons between the two groups since they're not mutually exclusive. You can't draw strong conclusions about the accuracy of records between the two groups without this kind of simple analysis. I think the paper would be a lot stronger.

New information shouldn't be introduced in the discussion. Move most of it to the introduction but refer back to it in the discussion so as to engage with the literature.

Specific comments:

pg 4, line 75 : What is the Final Rule of 2017?

pg 4, lines 92-96. "For a given drug, we examined whether each relevant trial listed in the FDA document was also registered with CTG. If so, we compared details of (i) phase, (ii) enrollment, and (iii) enrollment attribute (‘anticipated’ or ‘actual’) from the two sources. We did this for all trials, and separately for the sub-set of pivotal trials."

This should go in the methods and not the introduction.

I would also suggest introducing your research aim/question more explicitly, and also putting the 'why' in here (same paragraph)

pg 4 line 101: Please cite where the data is actually obtained from: not just 'publicly available information' but the actual links and names of the datasets. Also this should go in your data availability statement. This section duplicates some effort in the following subsection, so it may be sensible to cut it out entirely or revise to cite the datasets.

pg 4 line 102: Please go into slightly more detail about the structure of your discussions to resolve discrepancies.

pg 5 lines 105-107: How did you define what orphan drugs are? Was it your own definition of something from the FDA?

pg 5 lines 107-108: What details are in the database? A list of variables would be helpful here. (maybe in table form?)

pg 5 lines 113-115: How do you determine 'orphan indication'? How does this relate to 'orphan drugs'?

pg 5 lines 120-122: The 'single approval list' comes from the FDAOD database, I'm assuming? It's not clear here, so I would make it explicit which list is from which dataset.

pg 5 line 125: Why did you choose to study those approved after 2008? What's the rationale?

pg 6 line 134-136: The info provided in FDAOD should go with the information about the dataset itself.

pg 6 line 140: Indication here refers to whether or not the trial was for an orphan drug?

pg 6 lines 145-146: I'm not sure what you're trying to say here: what script did you use for data extraction, what exactly was extracted, and why then did you just extract it manually anyways (so you didn't use a script?)?

pg 6 lines 148-149: What does 'pivotal' mean in this context?

pg 7 lines 159-169: This paragraph is really difficult to follow due to all the acronyms. Some of these may need spelling out.

pg 7 line 180: What did you do with the remaining 220 studies?

pg 13 lines 243-245: Why did you limit the orphan indications? It is maybe because of 'the challenge of examining the MedRs' but I'm not sure why this is relevant to the research question. I'm not sure too that I would start on limitations in the discussion section - you have a separately limitations section on pg 15, lines 309-313 so move most of this here.

pg 14 lines 267-272: The data quality discussion needs to engage more with the literature on this. Why is it important that N1 and N2 match? Why should the reader care about these discrepancies? And why is 17% 'a large number'?

6. PLOS authors have the option to publish the peer review history of their article (what does this mean?). If published, this will include your full peer review and any attached files.

**Do you want your identity to be public for this peer review?** For information about this choice, including consent withdrawal, please see our Privacy Policy.

Reviewer #1: No

Reviewer #2: No

Reviewer #3: No

Reviewer #4: No

---

## [Editor Report · Decision Letter 1]

6 Dec 2021

PGPH-D-21-00617R1

Discrepancies between FDA documents and ClinicalTrials.gov for Orphan Drug-related clinical trial data

Dear Dr. Yubraj Achara,

Thank you for submitting your manuscript to PLOS Global Public Health. After careful consideration, we feel that it has merit but does not fully meet PLOS Global Public Health’s publication criteria as it currently stands. Therefore, we invite you to submit a revised version of the manuscript that addresses the points raised during the review process.

We look forward to receiving your revised manuscript.

Kind regards,

Genevieve Cecilia Aryeetey, Ph.D

Academic Editor
---

## [Decision Letter · Decision Letter 2]

3 Feb 2022

PGPH-D-21-00617R2

Discrepancies between FDA documents and ClinicalTrials.gov for Orphan Drug-related clinical trial data

Dear Dr. Gayatri Saberwal,

Thank you for submitting your manuscript to PLOS Global Public Health. After careful consideration, we feel that it has merit but does not fully meet PLOS Global Public Health’s publication criteria as it currently stands. Therefore, we invite you to submit a revised version of the manuscript that addresses the points raised during the review process.

We look forward to receiving your revised manuscript.

Kind regards,

Genevieve Cecilia Aryeetey, Ph.D

Academic Editor

Journal Requirements:

1. Please update your Competing Interests statement. If you have no competing interests to declare, please state: “The authors have declared that no competing interests exist.”

Additional Editor Comments (if provided):

Dear Authors,

We have received reviews from the revised manuscript. There are a few minor comments and suggestions from reviewer 1 which I would like the authors to address.

Reviewers' comments:

Reviewer's Responses to Questions

**Comments to the Author**

1. If the authors have adequately addressed your comments raised in a previous round of review and you feel that this manuscript is now acceptable for publication, you may indicate that here to bypass the “Comments to the Author” section, enter your conflict of interest statement in the “Confidential to Editor” section, and submit your "Accept" recommendation.

Reviewer #1: (No Response)

Reviewer #2: All comments have been addressed

Reviewer #3: All comments have been addressed

2. Does this manuscript meet PLOS Global Public Health’s publication criteria? Is the manuscript technically sound, and do the data support the conclusions? The manuscript must describe methodologically and ethically rigorous research with conclusions that are appropriately drawn based on the data presented.

Reviewer #1: Yes

Reviewer #2: Yes

Reviewer #3: Yes

3. Has the statistical analysis been performed appropriately and rigorously?

Reviewer #1: N/A

Reviewer #2: Yes

Reviewer #3: Yes

4. Have the authors made all data underlying the findings in their manuscript fully available (please refer to the Data Availability Statement at the start of the manuscript PDF file)?

Reviewer #1: Yes

Reviewer #2: Yes

Reviewer #3: Yes

5. Is the manuscript presented in an intelligible fashion and written in standard English?

Reviewer #1: Yes

Reviewer #2: Yes

Reviewer #3: Yes

6. Review Comments to the Author

Reviewer #1: The authors did a good job addressing the reviewers' comments. However there are some suggestions in the attached file that still need to be addressed.

Reviewer #2: The authors have addressed the major comments I had on the previous version of the manuscript. I would, therefore, like to recommend the manuscript for publication.

Reviewer #3: The authors have responded to the comments.

7. PLOS authors have the option to publish the peer review history of their article (what does this mean?). If published, this will include your full peer review and any attached files.

**Do you want your identity to be public for this peer review?** For information about this choice, including consent withdrawal, please see our Privacy Policy.

Reviewer #1: No

Reviewer #2: No

Reviewer #3: No

---

## [Editor Report · Decision Letter 3]

17 Feb 2022

Discrepancies between FDA documents and ClinicalTrials.gov for Orphan Drug-related clinical trial data

PGPH-D-21-00617R3

Dear Prof. Saberwal,

We are pleased to inform you that your manuscript 'Discrepancies between FDA documents and ClinicalTrials.gov for Orphan Drug-related clinical trial data' has been provisionally accepted for publication in PLOS Global Public Health.

Best regards,

Genevieve Cecilia Aryeetey, Ph.D

Academic Editor